# First Record of *Megamphopus katagani* Bakir, Sezgin & Myers, 2011 (Amphipoda, Photidae) in the Italian Waters: A Species Associated with the "Amphioxus Sand" Biocenosis

Emanuele Mancini [1,2,*], Giuseppe Catalano [3], Marco Lezzi [4], Francesco Tiralongo [2,5,6] and Andrea Bonifazi [7]

1. Italian Fishery Research and Studies Center, 00184 Rome, Italy
2. Ente Fauna Marina Mediterranea, Scientific Organization for Research and Conservation of Marine Biodiversity, 96012 Avola, Italy
3. Geonautics s.r.l., 92100 Agrigento, Italy
4. Struttura Oceanografica Daphne, ARPAE Emilia Romagna, 47042 Cesenatico, Italy
5. Department of Biological, Geological and Environmental Sciences, University of Catania, 95124 Catania, Italy
6. National Research Council, Institute of Marine Biological Resources and Biotechnologies, 60125 Ancona, Italy
7. Dipartimento Stato dell'Ambiente, ARPA Lazio, 00173 Rome, Italy
* Correspondence: mancini.e@cirspe.it

**Abstract:** The biocenosis of coarse sand and gravel exposed to bottom currents (SGCF) is the typical habitat of the amphioxus *Branchiostoma lanceolatum* (Pallas, 1774). This species and the habitat where it lives are threatened, and there are few reports of this biocenosis in the Italian and European waters. During a sampling survey carried out along the eastern Sicily coast (Ionian Sea) the macrozoobenthic community associated with this biocenosis was investigated. In this work, we report the presence of "Amphioxus sand" habitat in the Ionian Sicilian coast and the first record of the amphipod *Megamphopus katagani* Bakir, Sezgin and Myers, 2011 in the Italian waters.

**Keywords:** Mediterranean Sea; *Branchiostoma lanceolatum*; photidae; lancelet; SGCF biocenosis; macrozoobenthos





Soft bottoms are the most common habitats in coastal marine ecosystems, and the zonation of macrozoobenthic species are strictly related to the sediment characteristics and to their size distribution [1,2]. The "biocenosis" is the operational unit useful for classifying the different groups of benthic species according to the environmental characteristics that determine their distribution and their bathymetric zoning [3]. The biocenosis of coarse sand and gravel exposed to bottom currents (SGCF) *sensu* Pérès & Picard [3] is common both in high-energy infralittoral areas and in coastal environments with high deposition of coarse sediment and bioclast from adjacent seagrass and hard beds [1]. The SGCF biocenosis is the typical habitat of the amphioxus *Branchiostoma lanceolatum* (Pallas, 1774) (Figure 1) and usually hosts a rich and well-diversified macrozoobenthic community [3,4]. The distribution and local abundance of the lancelets are regulated by a combination of environmental factors comprising the characteristics of the sediments, the depth and the water temperature, while the sea currents mainly determine their distribution range [5]. To date, there are few reports of the "Amphioxus sand" in the European waters, and it has been found mainly in the Western Mediterranean and in the Adriatic Sea, while there is only one report both for the Black Sea and for the Eastern Mediterranean Basin [4,6,7]. Furthermore, the populations of *B. lanceolatum* and the SGCF-associated communities are threatened worldwide due to the increased human pressure in coastal marine environments (e.g., dredging, trawl fishing, pollution, eutrophication, etc.) [6,7].

In the "Amphioxus sand" biocenosis (Figure 2), we found the amphipod *Megamphopus katagani* Bakir, Sezgin and Myers, 2011 (Figures 3–8), a species belonging to the family Photidae that was recently described from the Sea of Marmara (Turkey), and it has never been reported in

the Italian waters. This amphipod was sampled on mixed sandy substrates in areas adjacent to coralligenous bioconstructions. Although the ecology of this species has not yet been described, the species belonging to the genus *Megamphosus* are considered deposit feeders that mainly feed on detrital organic matter [8]. *Megamphosus katagani* can be distinguished from the other congeneric species by the following diagnostic characteristics of the male: the coxa 2 is sub-square (Figure 5), and the dactylus of the gnathopod 1 is longer than propodus (Figure 6) [9]. It must be noted that antennae are lacking in all specimens as they break off easily during the collecting process. These are also not reported from the original description of the species.

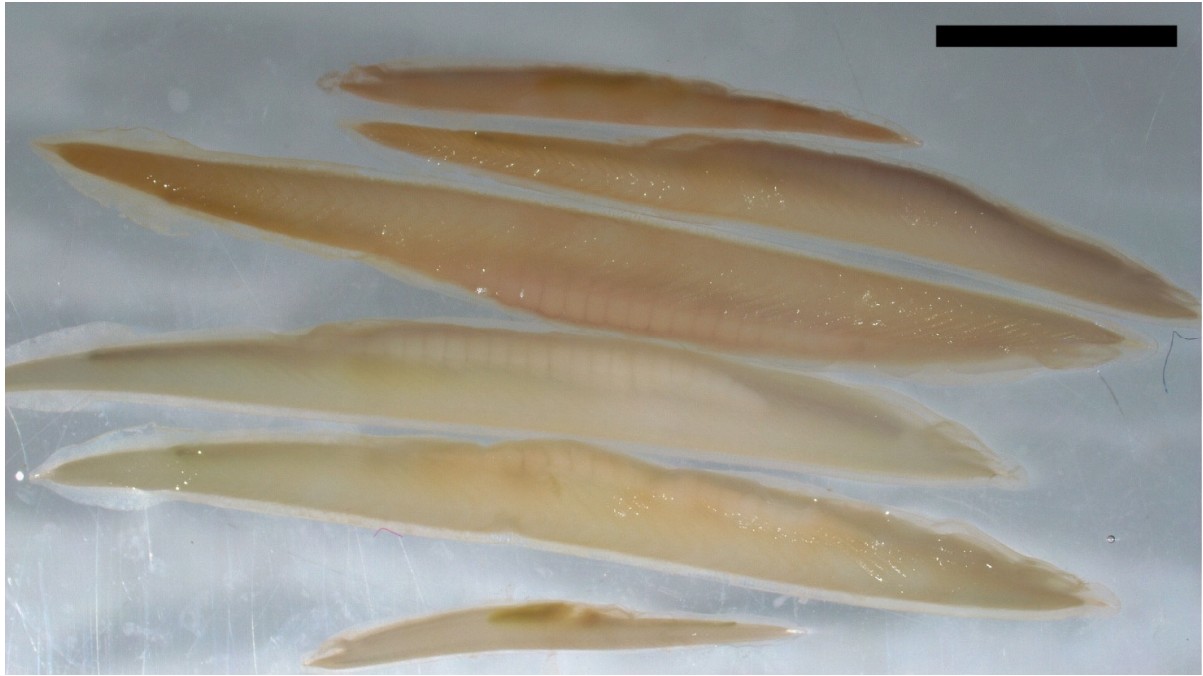

**Figure 1.** Some specimens of *Branchiostoma lanceolatum* sampled at Capo Mulini. Scale bar: 0.5 cm. Photo credit: Giuseppe Catalano.

Here we report the presence of the "Amphioxus sand" habitat in the Ionian Sicilian coast and the first record of *M. katagani* in the Italian waters. A total of 23 specimens of *M. kategani* and 25 individuals of *B. lanceolatum* were collected during a macrozoobenthic sampling survey carried out in the coastal area of Capo Mulini (37.57636 N, 15.17857 E) (Catania, Sicily, Italy), at the depth of 35 m. In the sampling site the seabed was characterized by the presence of pre-coralligenous and coralligenous biocostructions that stand out on gravelly coarse sand where the gravel fraction mainly consisted in volcanic gravels and bioclasts composed by skeletal fragments and shells (Figure 2). The original description of *M. kategani*, Bakir et al. [8] reported a similar habitat appropriate for this amphipod species. The samples were collected by using an 18 L Van Veen grab, and at each sampling, three replicates were collected and sieved with a mesh size of 1 mm. The retained fraction of sediment was preserved in 75% ethanol, and in the laboratory all organisms were sorted and identified to the finest taxonomic level possible and preserved in 75% ethanol. A total of 1429 individuals belonging 140 species were identified, and the analysis of the macrozoobenthic fauna associated with the "Amphioxus sand" habitat revealed that polychaetes were the dominant taxon, mostly represented by the serpulid *Ditrupa arietina* (O. F. Müller, 1776), the sabellid *Dialychone usticensis* (Giangrande, Licciano and Castriota, 2006) and the oweniid *Galathowenia oculata* (Zachs, 1923). The amphipods *Ampelisca brevicornis* (Costa, 1853) and *Urothoe intermedia* Bellan-Santini and Ruffo, 1986 and the sipunculan *Aspidosiphon (Aspidosiphon) muelleri muelleri* Diesing, 1851 were also well represented in the benthic community. Some species exclusive to the SGCF biocenosis [3], such as the ophiuroid *Ophiopsila annulosa* (M. Sars, 1859), the bivalve molluscs *Glycymeris glycymeris* (Linnaeus,

1758) and *Dosinia exolete* (Linnaeus, 1758), the polychaete worm *Scoloplos typicus* (Eisig, 1914) and the decapod *Anapagurus breviaculeatus* Fenizia, 1937 were also found; moreover, among the exclusive preferential species were the polychaete worm *Euthalenessa oculata* (Peters, 1854) and the bivalve mollusc *Gari costulata* (W. Turton, 1822).

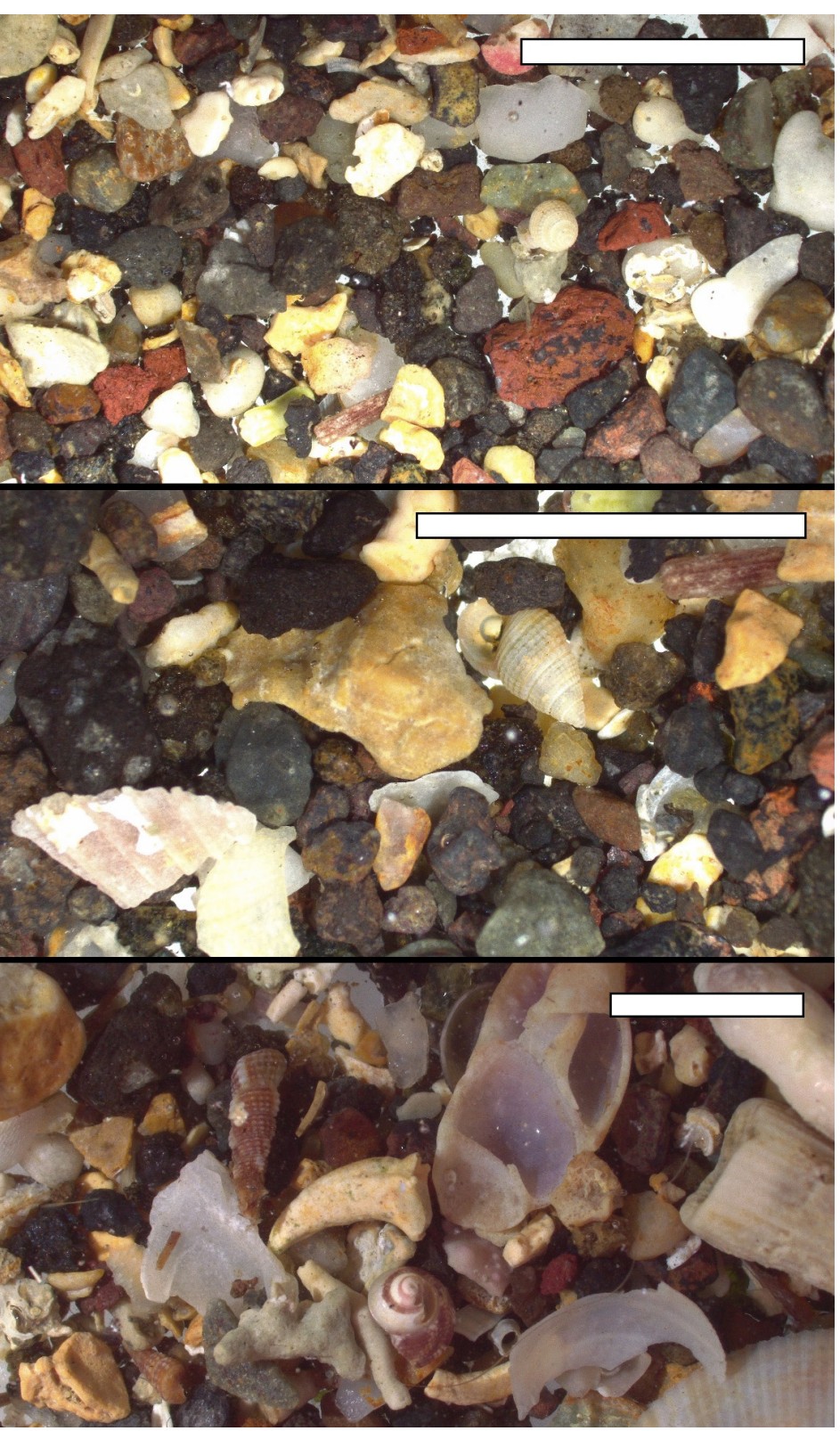

**Figure 2.** Close up of gravel fraction of the "Amphioxus sand". Scale bar: 1 cm. Photo credit: Giuseppe Catalano.

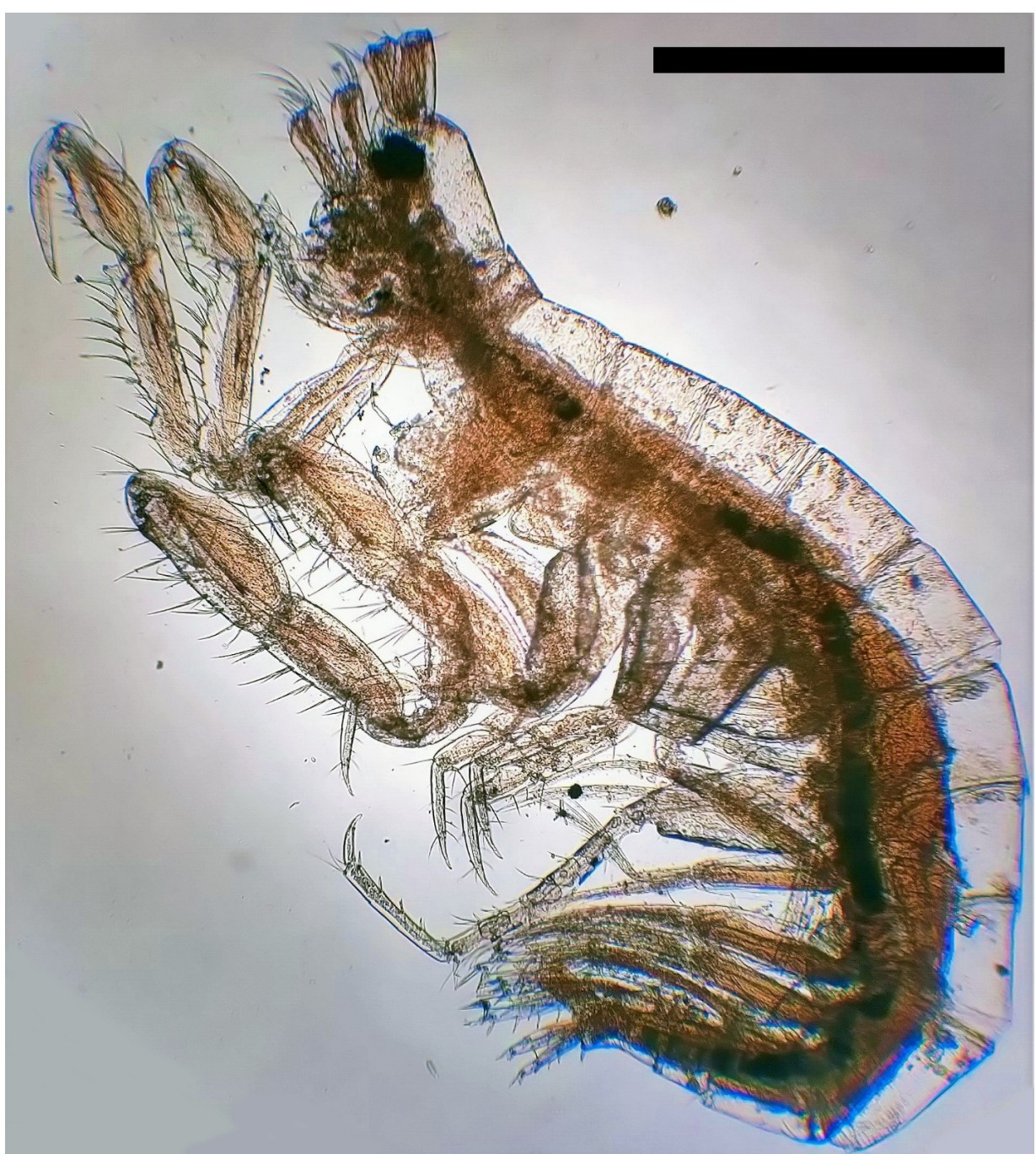

**Figure 3.** *Megamphosus katagani*: lateral view of a male specimen (antennae broken off). Scale bar: 1 mm. Photo credit: Emanuele Mancini.

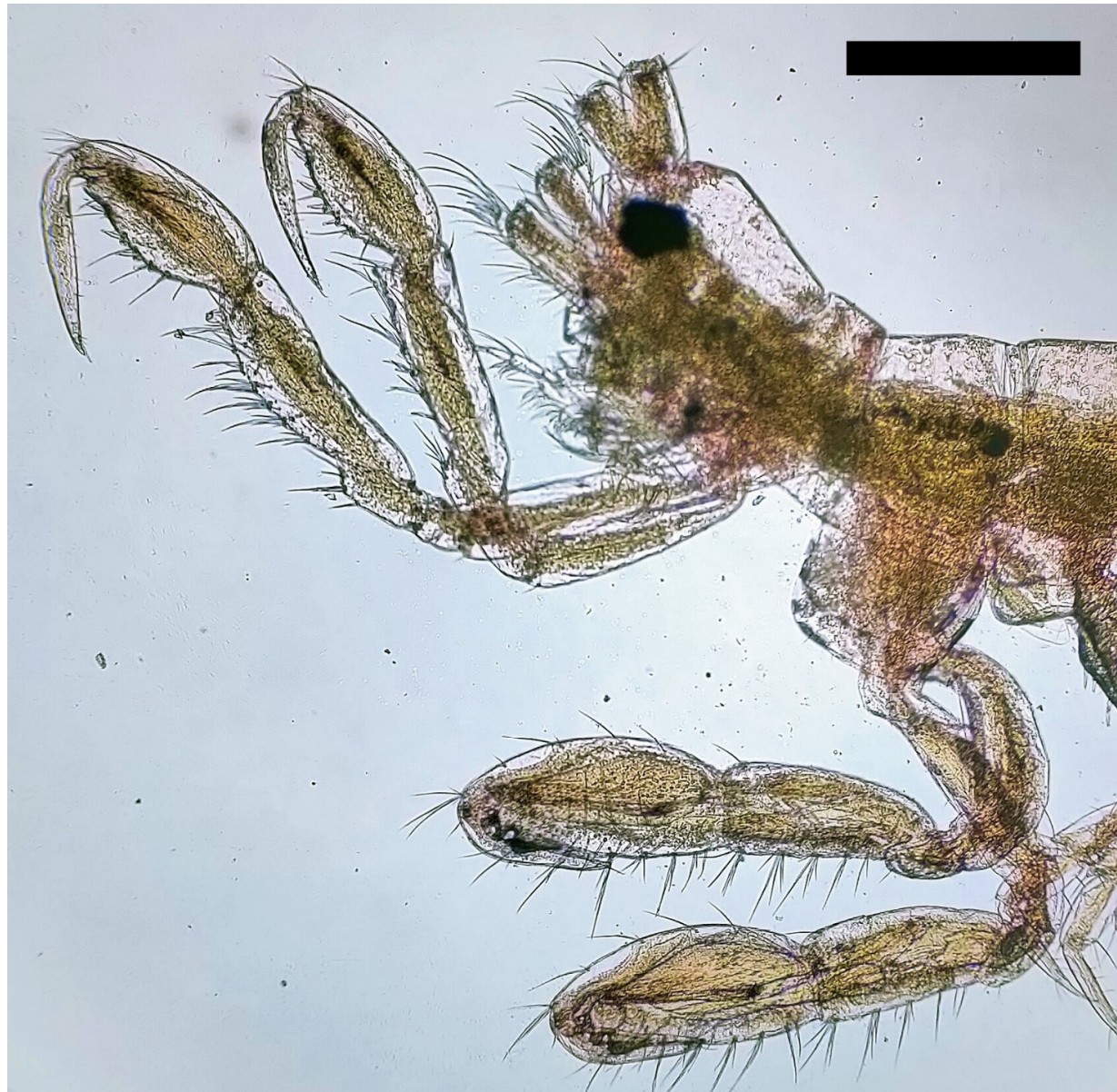

**Figure 4.** *Megamphosus katagani*: lateral view of anterior region (antennae broken off). Scale bar: 0.5 mm. Photo credit: Emanuele Mancini.

This work increases the knowledge concerning the distribution of the "Amphioxus sand" habitat in the Mediterranean Sea and constitutes both the second record worldwide for the amphipod *M. katagani* and the first record of this species in the Italian waters, providing new insights about the ecology of this species. Furthermore, we provide the photographic material useful for the identification of this species. It is noteworthy that the high number of species found in association with the SGCF highlights the role of this biocenosis as biodiversity hotspot.

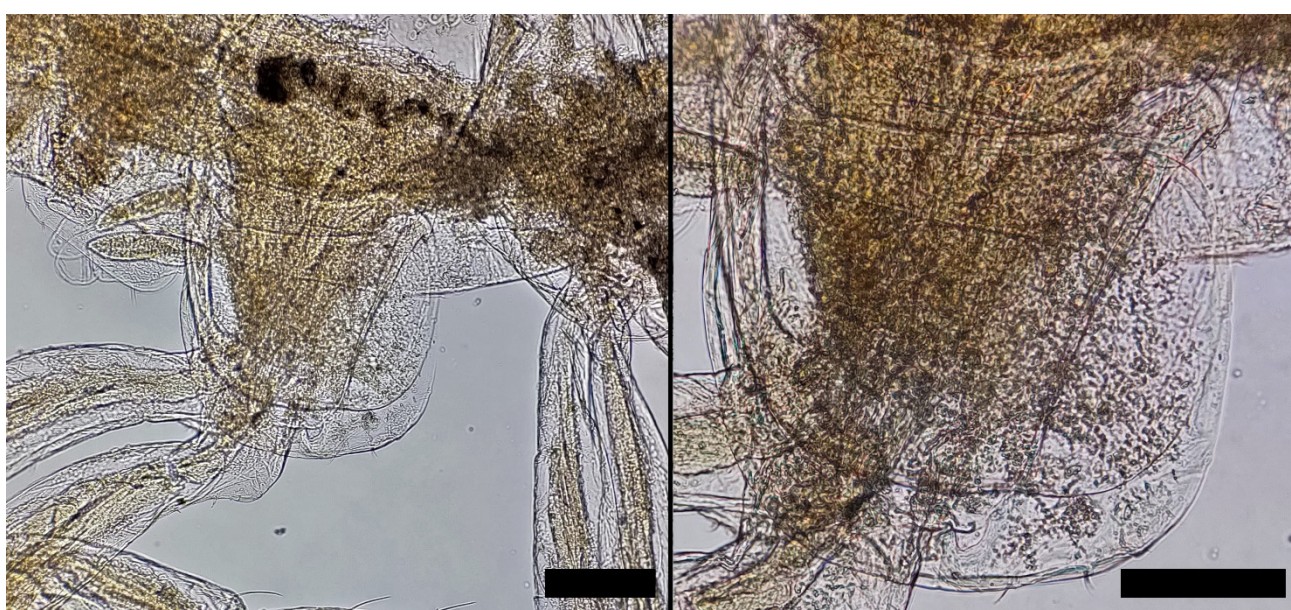

**Figure 5.** *Megamphosus katagani*: coxa 2 at different levels of magnification. Scale bar: 0.1 mm. Photo credit: Emanuele Mancini.

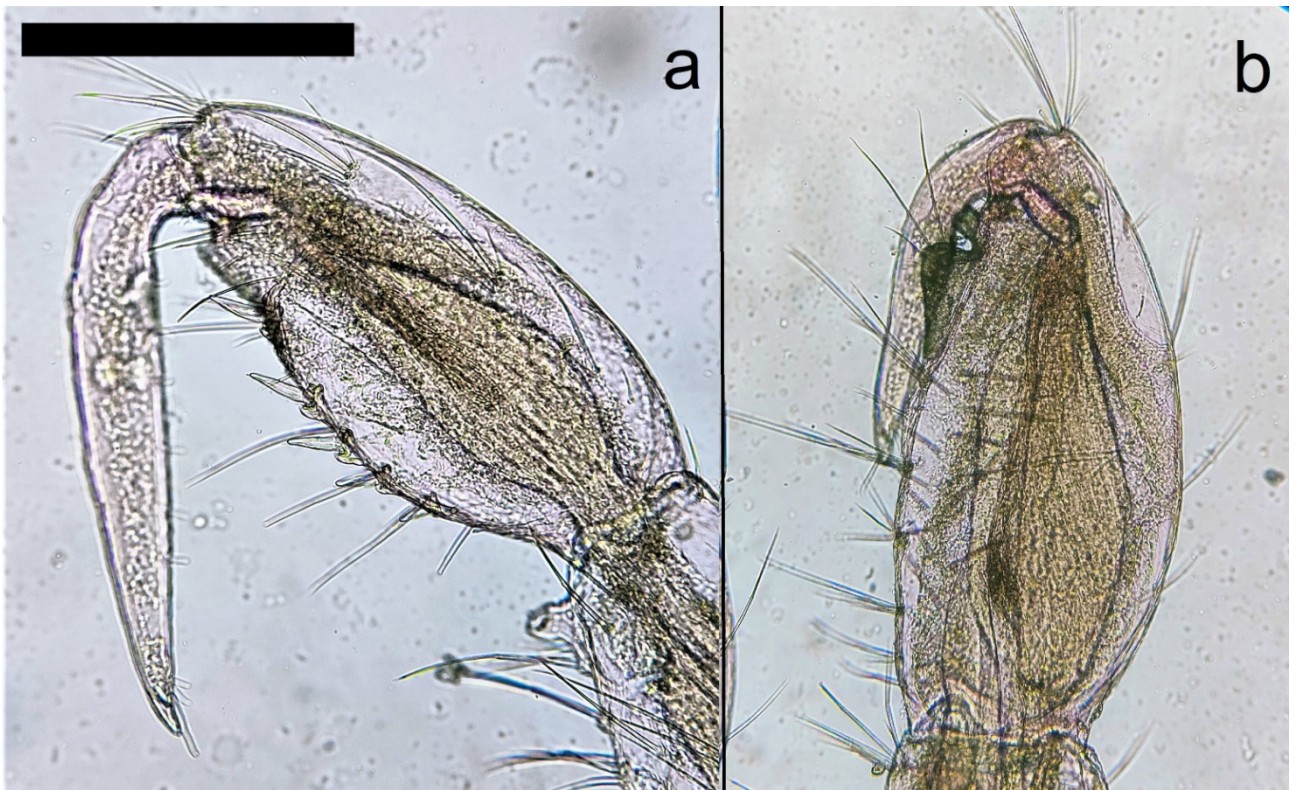

**Figure 6.** *Megamphosus katagani*: gnathopod 1 (**a**) and gnathopod 2 (**b**). Scale bar for all the pictures: 0.25 mm. Photo credit: Emanuele Mancini.

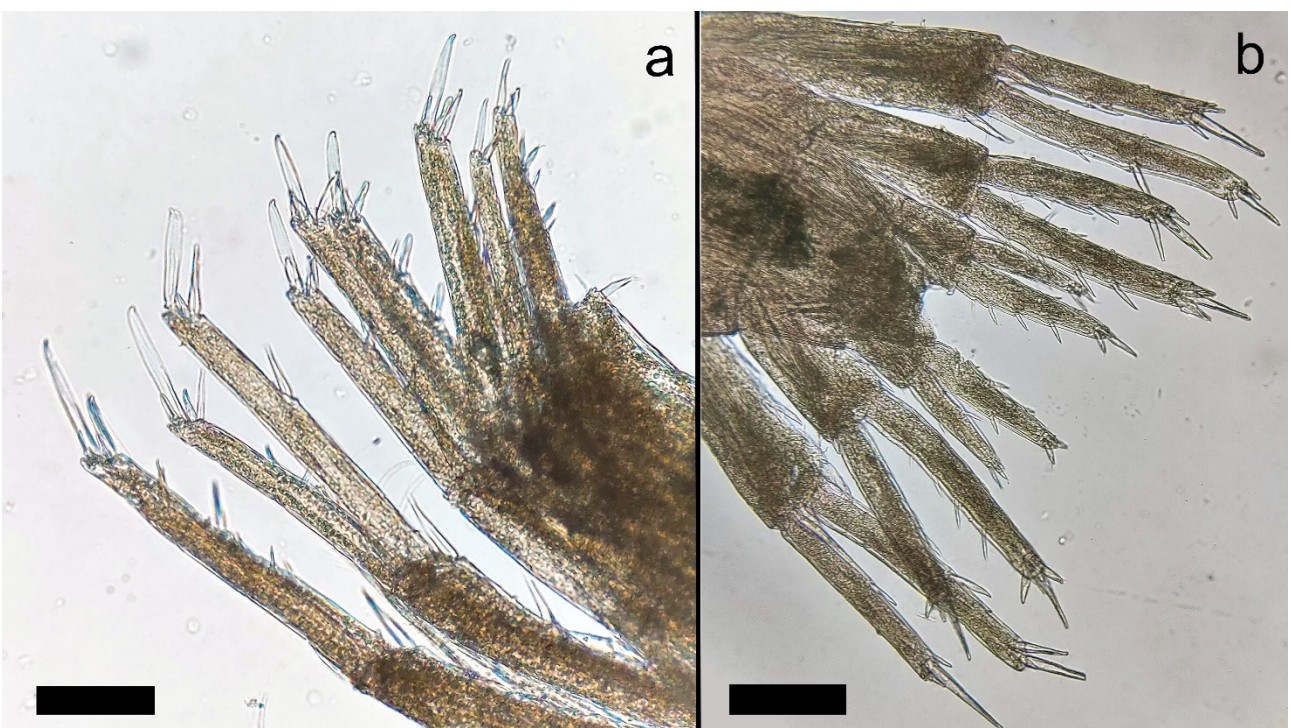

**Figure 7.** *Megamphosus katagani*: (**a**) lateral view of urosome (**b**). dorsal view of urosome. Scale bar for all the pictures: 0.1 mm. Photo credit: Emanuele Mancini.

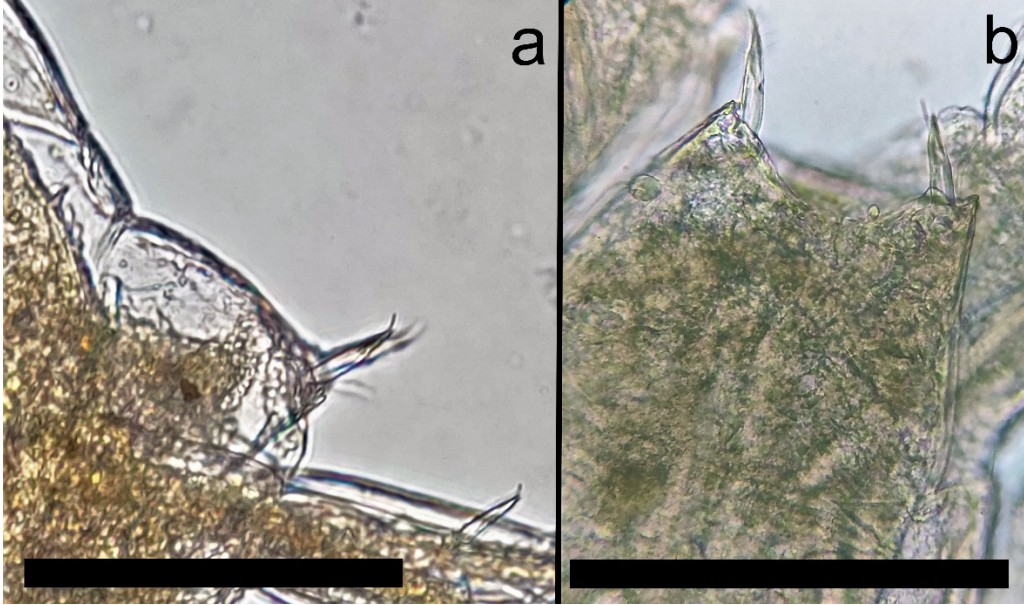

**Figure 8.** *Megamphosus katagani*: (**a**) lateral view of telson (**b**). dorsal view of telson. Scale bar for all the pictures: 0.5 mm. Photo credit: Emanuele Mancini.

**Author Contributions:** E.M. analyzed and described the specimens and wrote the manuscript. A.B. analyzed, described, and photographed (stereo and optical microscope) the specimens and wrote the manuscript. G.C. collected the specimens and revised the manuscript. M.L. examined the specimens and revised the manuscript. F.T. wrote and revised the manuscript. All authors have read and agreed to the published version of the manuscript.

**Funding:** This research received no external funding.

**Institutional Review Board Statement:** Not applicable.

**Data Availability Statement:** Not applicable.

**Conflicts of Interest:** The authors declare no conflict of interests.

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
