# Peer review of "First Record of Megamphopus katagani Bakir, Sezgin & Myers, 2011 (Amphipoda, Photidae) in the Italian Waters: A Species Associated with the “Amphioxus Sand” Biocenosis"

_diversity, doi:10.3390/d15030358_

Round 1

Reviewer 1 Report

Interesting contribution with good quality photo's of an amphipod. I made some minor remarks about grammar in the pdf.  A scale bar with the photo's of gravel fraction might help to easily estimate the coarseness of the sand/gravel nature of the habitat for a reader.

Author Response

Thank you for your corrections and for your valuable comments which will increase the quality of the manuscript. We have modified and corrected the manuscript following your instructions, in details:

  • We have added the scale bar in the Fig.1;
  • the sentence relating to the lack of antennae in the specimens has been added to the text;
  • we have improved the English and corrected the grammar following your instructions;
  • the captions of the Figg. 2 and 3 were modified.

Sincerely,

Emanuele Mancini

Reviewer 2 Report

This MS describe the first record of the amphipod Megamphopus katagani in the amphioxus sand in Italy. The authors would need to elaborate the geographical distribution of M. katagani (where type locality is in Turkey in 2011), where in the Mediterranean has this species recorded? Have they recorded in many locations in the Mediterranean. If so, having one more locations seems did not get very great contribution to science. I would suggest the author to add in more information for example the trophic ecology of this species if it is associated with amphioxus sands. How common is amphioxus sands in the Mediterranean?

For introducing amphioxus sand and amphioxus distribution mentioned in the first paragraph, I would suggest the author to cite the reference below, which shows the amphioxus distribution is affected by sediment characterisitcs (amphioxus sand), water temperature and depth of water. 

Lin, C.-H., Chen, J.P., Chan, B.K.K. and Shao, K.T. (2014). The interplay of sediment characteristics, depth, water temperature, and ocean currents shaping the biogeography of lancelets (Subphylum Cephalochordata) in the NW Pacific waters. Marine Ecology 2014:1-14, doi: 10.1111/maec.12183

Author Response

Thank you for your corrections and for your valuable comments which will increase the quality of the manuscript. We have modified and corrected the manuscript following your instructions, in details:

  • Regarding your comment “The authors would need to elaborate the geographical distribution of katagani (where type locality is in Turkey in 2011), where in the Mediterranean has this species recorded? Have they recorded in many locations in the Mediterranean. If so, having one more locations seems did not get very great contribution to science” we would like to point out that our work represents the second report of this species worldwide; to date, M. katagani has been reported exclusively in the Turkish waters. To make this clearer, a sentence has been added in the conclusions “This work increases the knowledge concerning the distribution of the “Amphioxus sand” habitat in the Mediterranean Sea and constitutes both the second record worldwide for the amphipod M. katagani and the first record of this species in the Italian waters”;
  • as suggested, some information regarding the trophic ecology of the genus Megamphopus has been added;
  • regarding your comment “How common is amphioxus sands in the Mediterranean?” we have inserted a sentence to increase the information on its distribution in the Mediterranean basin;
  • as you suggested, we have inserted the citation of Lin et al.,2015, and we have also added a sentence concerning the environmental characteristics that determine the distribution of lanceolatum.

Sincerely,

Emanuele Mancini